# Utilizing Pork Exudate Metabolomics to Reveal the Impact of Aging on Meat Quality

**DOI:** 10.3390/foods10030668

**Published:** 2021-03-20

**Authors:** Qianqian Yu, Bruce Cooper, Tiago Sobreira, Yuan H. Brad Kim

**Affiliations:** 1Meat Science and Muscle Biology Laboratory, Department of Animal Science, Purdue University, West Lafayette, IN 47906, USA; yuqianqianlly@hotmail.com; 2College of Life Science, Yantai University, No. 30 Qingquan Road, Laishan District, Yantai 264005, China; 3Bindley Bioscience Center, Purdue University, West Lafayette, IN 47907, USA; brcooper@purdue.edu (B.C.); sobreira@purdue.edu (T.S.)

**Keywords:** pork purge, aging, metabolome, quadrupole time-of-flight mass spectrometer (UHPLC-QTOF-MS)

## Abstract

This study was performed to assess the changes in meat quality and metabolome profiles of meat exudate during postmortem aging. At 24 h postmortem, longissimus lumborum muscles were collected from 10 pork carcasses, cut into three sections, and randomly assigned to three aging period groups (2, 9, and 16 d). Meat quality and chemical analyses, along with the metabolomics of meat exudates using ultra-high-performance liquid chromatography coupled with a quadrupole time-of-flight mass spectrometer (UHPLC-QTOF-MS) platform, were conducted. Results indicated a declined (*p* < 0.05) display color stability, and increased (*p* < 0.05) purge loss, meat tenderness, and lipid oxidation as aging extended. The principal component analysis and hierarchical clustering analysis exhibited distinct clusters of the exudate metabolome of each aging treatment. A total of 39 significantly changed features were tentatively identified via matching them to METLIN database according to their MS/MS information. Some of those features are associated with adenosine triphosphate metabolism (creatine and hypoxanthine), antioxidation (oxidized glutathione and carnosine), and proteolysis (dipeptides and tripeptides). The findings provide valuable information that reflects the meat quality’s attributes and could be used as a source of potential biomarkers for predicting aging times and meat quality changes.

## 1. Introduction

Postmortem aging is a value-adding process and has been extensively practiced by the global meat industry for years to enhance meat tenderness, juiciness, and flavor development [1]. In particular, wet-aging (storing fresh meat at refrigerated temperatures under vacuum packaged bags) is a widely used strategy in the meat industry due to its enhanced ease and flexibility of storage [2]. However, an increase in exudate from meat during aging in a vacuum bag is unavoidable [3,4]. The main cause for the exudate release is a series of breakdowns in the myofibrils, the cell membrane structure, and the intracellular cytoskeleton during aging process [5]. In addition, physical pressure applied during vacuum packing causes the extraction of liquid from the meat, resulting in a greater release of exudation and purge collection in the bag [6]. The purge contains numerous metabolites, which could provide useful background biochemical information and could potentially be closely associated with meat quality changes. Purge, however, is mostly considered to be waste, and the potential of meat purge use for meat quality determination has not been actively realized. Only a few studies have explored the potential value of meat purge as an analytical medium.

Meat purge is the aqueous solution that contains water-soluble sarcoplasmic proteins, nucleotides, amino acids, peptides, and many soluble enzymes, which are all necessary for the metabolism of muscle and for postmortem changes in meat [7,8]. Research regarding purge loss has mainly focused on identifying exudation amount or correlating its relationship with the meat’s capacity to hold water, meat tenderness, or protein oxidation [9,10,11]. Very limited research has been done regarding the complete profiling of the compounds in meat exudate itself. Recently, the metabolomes in beef [7] and pork [12] exudates during different aging periods were first characterized by ^1^H nuclear magnetic resonance (NMR) spectroscopy, and the results indicated a strong correlation between exudate and meat spectra. Moreover, the metabolite profiles of the muscle exudate obtained from chicken breast fillets affected by wooden breast (WB) myopathy were revealed by NMR-based metabolomics (eleven metabolites were significantly affected by WB myopathy) [13]. These researches display valuable information about the metabolome profile changes of meat exudate and suggest an easy method to monitor metabolic profile changes associated with meat aging or meat quality by determining exudate spectra.

Mass spectrometry (MS) has emerged as the critical technology in metabolomics studies due to its unparalleled sensitivity and specificity, high resolution and wide dynamic range, enabling the comprehensive quantitative and qualitative measurement of large-scale small-molecular metabolites in complex biological samples [14]. It has been widely applied to characterize and quantify the component changes in various muscles [15,16,17,18] during postmortem aging. However, most published studies only focus on muscle tissue rather than its purge solution. In our previous study, metabolites present in exudates from beef loins and tenderloins at different aging periods (9, 16, and 23 d) were characterized using ultra-high-performance liquid chromatography coupled with electrospray ionization mass spectrometer (UPLC-ESI-MS), and the relationship between identified metabolites and the oxidative stability of beef muscles during aging was examined as well [19]. The results indicated that both the oxidative stability of meat and well as the metabolomics profile of meat exudate were significantly affected by muscle type and aging period. Furthermore, some oxidative stability-related compounds in meat exudate were identified, which could possibly explain changes in meat quality.

Taken together, the objective of the study was to assess the meat quality attribute changes and the associated purge metabolome profiles during vacuum (wet) aging in order to characterize the major metabolites that present in pork exudate with different aging periods for monitoring aging times and potentially determining changes in meat quality.

## 2. Materials and Methods

### 2.1. Sample Collection

A total of 10 pork carcasses (Landrace × Large White breed with an average live weight of 117.3 ± 1.7 kg) were harvested at the Purdue University Meat Laboratory. Longissimus lumborum muscles from one side of each carcass were collected at 24 h postmortem, then split in three sections, vacuum packaged, and randomly assigned to three aging times (2, 9, and 16 days) at 2 °C. Upon the completion of each aging period, the vacuum packages were opened and the exudates were collected and stored at −80 °C until used for metabolomics analysis. The weight of each section before and after aging were recorded in order to calculate purge loss. Pork sections were divided into multiple cuts, and two cuts (around 2.54 cm thick) of each section were used for shear force and display color evaluation, respectively. The rest of the cuts were vacuum packaged individually and stored at −80 °C until chemical analyses were conducted.

### 2.2. pH Measurement

The pH of the pork samples was measured at 24 h postmortem and after each assigned aging time using a meat pH probe (HANNA HI 99163, Hanna Instrument, Inc., Warner, NH, USA). The pH probe was calibrated to a pH 4 and 7 buffer at 2 °C before use and was directly inserted into three random locations of each sample for measurement.

### 2.3. Water-Holding Capacity (WHC)

Multiple measurements (including purge loss, drip loss, and cook loss) were conducted to determine the weight differences under various physical stressors to evaluate the WHC of the pork samples. Purge loss was measured and calculated in accordance to Setyabrata et al. [20]. Drip loss was performed according to the method obtained from Kim et al. [21]. To evaluate cooking loss, the pork cuts were first weighed and then cooked at 135 °C on a flat top electric griddle (Farberware, Walter Idde and Co., Bronx, NY, USA). Each cut was turned to its other side when the core temperature reached 41 °C and then cooked to 71 °C. After cooking, the sample’s weight was recorded and the cooking loss was calculated according the following formula:$$\mathrm{Cooking}\;\mathrm{loss}\;\left(\%\right)=100\times \left(1-\frac{weight\;of\;cooked\;sample}{weight\;of\;raw\;sample\;}\right)$$

### 2.4. Display Color

One cut from each assigned aging section was used for display color assessment. Briefly, pork cuts were aerobically packaged with polyvinyl chloride film (23,000 cm^3^/m^2^/24 h oxygen transmission rate at 23 °C) and displayed for 5 days at 2 °C under fluorescent light (∼1450 lx; color temperature = 3500 K). Color was recorded daily in five random locations on the pork cuts using a Minolta CR-400 colorimeter (Konica Minolta Photo Imaging Inc., Tokyo, Japan) with CIE standard illuminant D65. The chroma and hue angle were calculated according to the following formulae described by the American Meat Science Association guidelines [22]:$$\;chroma=\sqrt{\left(a^{*2}+b^{*2}\right)}\;\mathrm{hue}\;\mathrm{angle}=\arctan\left(b^{*}/a^{*}\right)$$

### 2.5. Shear Force

The cooked pork cuts were used for shear force evaluation after 24 h of cooling at 4 °C. Six cores parallel to the fiber direction were collected using a 1.27 cm diameter hand-held coring device. The measurement was performed utilizing a TA-XT Plus Texture Analyzer (Stable Micro System Ltd., Godalming, UK) coupled with a Warner–Bratzler shear attachment. The average peak shear force (Kg) from the cores was calculated for analysis.

### 2.6. Myofibril Fragmentation Index (MFI)

MFI was performed in accordance with Culler et al. [23], with some modifications. Two grams of the minced sample were homogenized with 20 mL cold (2 °C) MFI buffer (100 mM potassium chloride, 20 mM potassium phosphate, 1 mM ethylene glycol tetraacetic acid, 1 mM magnesium chloride, and 1 mM sodium azide; pH 7.0) for 45 s. The mixture was centrifuged at 1000× *g* for 15 min (at 4 °C) and then the supernatant was removed. The pellet was resuspended in 20 mL MFI buffer by using a stir rod and centrifuged again at 1000× *g* for 15 min. The supernatant was removed and a pellet of it was resuspended using 5 mL of cold (2 °C) MFI buffer. Straining sample using a No. 18 polyethylene strainer and an additional 5 mL buffer were used to facilitate the passage of myofibrils through the strainer. The protein concentration of the suspension was diluted with an MFI buffer to a final volume of 0.5 mg/mL and the absorbance was then measured using a UV spectrophotometer (VWR UV-1600 PC, VWR International, San Francisco, CA, USA) at 540 nm. The MFI was expressed as the absorbance multiplied by 200.

### 2.7. Lipid Oxidation

Lipid oxidation was evaluated via 2-thiobarbituric reactive substances (TBARS) assay according to the method outlined by Setyabrata and Kim [20]. Briefly, 5 g meat samples were homogenized with 15 mL of deionized distilled water and 50 μL of 10% butylated hydroxyl anisole solution in 90% ethanol for 30 s at 6000 rpm. One milliliter of the homogenate was mixed with 2 mL of 20 mM 2-thiobarbituric acid solution in 15% trichloroacetic acid, heated in 80 °C for 15 min, and cooled in ice for 10 min. Afterwards, the mixture was centrifuged at 2000× *g* for 10 min and filtered. The absorbance of the filtrate was read at 531 nm using a microplate spectrophotometer (BioTek Instruments, Inc., Winooski, VT, USA) and multiplied by 5.54 to express the TBARS value.

### 2.8. Protein Carbonyl Content

The carbonyl content of the pork samples was obtained through following a procedure described by Vossen and De Smet [24]. The carbonyl content was determined via derivatization with 2,4-dinitrophenylhydrazine (DNPH), leading to the formation of stable dinitrophenyl (DNP) hydrazone adducts, which can be detected spectrophotometrically at 375 nm using the Epoch Spectrophotometer System (BioTek Instruments, Inc., Winooski, VT, USA). The carbonyl concentration was expressed as nmol/mg protein.

### 2.9. Metabolomics Analysis for Pork Purge

#### 2.9.1. Metabolite Extraction

Purge metabolite extraction was performed according to the method from Bligh and Dyer [25]. Chloroform (250 μL) and methanol (225 μL) were added to 100 μL of meat purge, and then the samples were vortexed for 15 s and left to sit for 5 min. Afterwards, 125 μL of water was added, mixed, and centrifuged at 16,000× *g* for 8 min. The upper layer was transferred to vials and dried via a termovap sample concentrator. For ultra-high-performance liquid chromatography coupled with a quadrupole time-of-flight mass spectrometer (UHPLC-QTOF-MS) analysis, the dried fraction was redissolved in 100 μL of acetonitrile aqueous solution (water:acetonitrile = 95:5) containing 0.1% formic acid.

#### 2.9.2. UHPLC-QTOF-MS Analysis

An Agilent 1290 Infinity II UHPLC system (Agilent Technologies, Palo Alto, CA, USA), coupled with a Waters Acquity HSS T3 (2.1 × 100 mm × 1.8 um) separation column (Waters, Milford, MA, USA) and a HSS T3 (2.1 × 5 mm × 1.8 um) guard column, was used to obtain the chromatographic separations with the column temperature set at 40 °C. The sample injection volume was 5 μL. The mobile phases consisted of solvents A (0.1% formic acid in ddH_2_O) and B (0.1% formic acid in acetonitrile) with a 0.45 mL/min flow rate. The gradient program was as follows: 0–1 min, 100% A; 1–16 min, a linear gradient to 70% A; 16–21 min, a linear gradient to 5% A, and hold for 1.5 min; returning to 100% A over 1 min, and hold for 5 min for column re-equilibration. An Agilent 6545 quadrupole time-of-flight (Q-TOF) mass spectrometer was used for mass spectrometric analysis. The positive electrospray ionization (ESI) mode was applied for mass spectral (70–1000 *m/z*) data collection. The ESI capillary voltage was 3.5 kV, the drying gas flow rate was 8.0 L/min, the temperature of the nitrogen gas was 325 °C, the nebulizer gas pressure was 30 psig, the fragmentor voltage was 130 V, the skimmer was 45 V, and the Oct 1 RF Vpp was 750 V. Mass data were acquired using Agilent MassHunter B.06.

#### 2.9.3. Metabolomics Data Processing

The raw data were preprocessed (undergoing peak detection, deconvolution, and alignment) using Agilent ProFinder (v B.08). The peak area matrix was imported to MetaboAnalyst 4.0 for principal component analysis (PCA) and hierarchical cluster analysis (HCA). Variables with missing values were dealt a small value (half of the minimum values in the original data). The data were normalized by the sum, transformed by generalized log transformation, and auto scaled. Pairwise comparisons (16 d versus 2 d, 9 d versus 2 d, and 16 d versus 9 d) were performed, respectively, and features with a minimum fold change of 2 (ratio > 2 or <0.5; FDR < 0.05) were considered to have been changed significantly in the three comparison groups. Identifications were aided by performing data-dependent MS/MS collection on composite samples with 10 eV, 20 eV, and 40 eV of collision energy, respectively. Annotation was achieved based on the database of METLIN (www.metlin.scripps.edu (accessed on 20 January 2021)), with a mass tolerance of 10 ppm for MS1 and a mass tolerance of 20 ppm for MS2.

### 2.10. Statistical Analysis

The experimental design of this study was a completely randomized design. Each loin was used as an experimental unit. A repeated measure design was used to detect instrumental color parameters at three aging periods of retail display. The data of meat quality attributes were analyzed using SPSS (SPAA Inc., Chicago, IL, USA) for one-way ANOVA to evaluate the significance of the main treatment (aging). Significance was defined at the level of *p* < 0.05.

## 3. Results and Discussion

### 3.1. pH

The pH of the muscles was significantly affected by aging (Table 1). At 24 h postmortem, the ultimate pH dropped to 5.65 due to the accumulation of lactate and hydrogen ions from postmortem glycolysis and ATP hydrolysis [26]. The pH declined continually to 5.55 after 2 d of aging, and increased (*p* < 0.05) to 5.59 and 5.57 after 9 d and 16 d of aging, respectively. The increased pH may be due to the changes in charge caused by proteolytic enzymes during postmortem aging [27].

### 3.2. WHC

Purge loss increased from 3.23% at 2 d postmortem to 7.80% at 16 d postmortem (*p* < 0.05) (Table 1). The increase in purge loss with aging could be due to the degradation of proteins by proteolysis during aging, resulting in a breakdown in the myofibrils, cell membrane structure, and intracellular cytoskeleton, which induces the migration of myowater more easily from the intramyofibrilar space [5].

Drip loss dramatically decreased from 6.17% to 1.46% with aging (*p* < 0.05) (Table 1). Higher drip loss has been shown to occur during first 2 days postmortem, which corresponds to the findings from Straadt et al. [28], wherein drip loss increased from 24 h to 48 h postmortem and stayed constant until 4 d postmortem, after which point it decreased up until 14 d postmortem. From one side, the decreased drip loss should indicate the increased WHC of meat during aging according to the hypothesis of Kristensen and Purslow [29], which showed that degradations in the cytoskeleton during aging reduce or remove the linkage between the rigor-induced lateral shrinkage of myofibrils and the shrinkage of the whole muscle fiber. Therefore, muscle myofibrils are observed to “swell” and are capable of holding more water during aging [28]. However, an increased purge loss from meat during aging in a vacuum bag was still observed (Table 1), meaning that more water was exuded from meat, which may result in a lesser volume of water in the meat that was available to be released as drip loss.

As for cooking loss, no significant differences among aging periods were found, although 16 d postmortem showed a lower value numerically (Table 1). The weight loss of the meat during cooking is related to how much water is present and how easily it can leave the muscle structure network [5]. Purge loss increased following aging, indicating a lesser volume of water available in the muscle to be easily evaporated. Thus, no significantly changed cooking loss may suggest a decreased water-holding capacity as aging progresses.

### 3.3. Color Stability

The effect of wet-aging on the instrumental color parameters of the pork samples during display (0, 1, 2, 3, 4, and 5 days) is reported in Figure 1. Lightness (L*) was affected by aging time; 2 d aged meat showed lower values than did 9 d and 16 d aged meat at the first 3-day display. Additionally, L* development showed a significant increase (*p* < 0.01) in 2 d aged meat during display periods. Pork cuts from 16 d aged sections showed higher (*p* < 0.01) a* values than did 2 d and 9 d aged samples at the first 2-day display. However, at the end of the display period, 2 d aged cuts presented significantly higher (*p* < 0.05) a* values than did 9 d and 16 d aged ones. Moreover, a* values were all affected by display (*p* < 0.01) regardless of aging treatments and showed a significantly decreased tendency from that exhibited during day 1 display. Chroma, which represents the color intensity of the meat, was also affected by aging time (*p* < 0.01). Compared with shorter aging treatments, 16 d aged cuts showed the highest chroma at the first 2-day display. However, they presented more significant declines and showed the lowest chroma at the end of display. The decrease in chroma from 1 to 5 days of display indicates a gradually increased accumulation of surface metmyoglobin. These results may suggest that longer aging treatment could reduce meat color stability during display period [30]. Aged meat blooms more rapidly but subsequently browns earlier than fresh meat [31]. Previous research also indicated that the lower color stability of aged meat compared with unaged meat becomes evident after 3 days of display [32].

### 3.4. Shear Force and MFI

The shear force values of the pork samples were affected (*p* < 0.05) by aging time (Figure 2a). Two-day aged samples showed the highest shear force value, and the shear force value decreased as the aging period extended, which suggested an increased tenderness of the meat. At the same time, MFI significantly increased (*p* < 0.05) from 80.92 after 2 d postmortem aging to 110.76 with 9 d aging, and no significant difference was found between 9 d and 16 d aging treatments (Figure 2b). A higher MFI indicates a more severe myofibrillar degradation and thus a subsequently more tender meat [33]. Aging improves the tenderness of the meat through disruption of the muscle ultrastructure by intracellular proteolytic systems (including calpains, the lysosomal proteases, and cathepsins), which have the capability of degrading myofibrillar and cytoskeletal proteins [34].

### 3.5. Oxidative Stability

TBARS was performed to evaluate the lipid oxidation of the pork meat during aging. As showed in Figure 3a, there was a significant increase (*p* < 0.05) in the TBARS values of pork samples after 9 d aging than that of 2 d aged samples; however, no significant difference was found between 9 d and 16 d aging treatments. As for protein oxidation, carbonyl content increased numerically during aging even though no statistical significance was found (Figure 3b). Lipid oxidation is a critical deterioration reaction in meat and is positively correlated with oxymyoglobin oxidation [35]. During postmortem aging, muscle cells gradually lose their ability to maintain reducing conditions due to the collapse of the antioxidant system and structural integrity, which may lead postmortem muscles to be more susceptible to oxidation. Similarly, 14-day vacuum aged beef muscle indicated a higher lipid oxidation than that in nonvacuum packaged meat, as described by Popova et al. [36]. However, some researchers reported no aging effect on the initial TBARS from unaged or wet aged beef for 14 days [37] or no significant change in the TBARS of beef during aging for 14 days [38].

### 3.6. Metabolomics Profiling of Pork Exudate

A total of 1483 features were detected via UHPLC-QTOF-MS in the 15 purge samples (5 biological replicates × 3 aging treatments), and 20 of them were deleted due to having more than 50% of their values missing. Thus, a total of 1463 features were used for PCA and HCA analyses. There were complete separations among three different aging treatments from the PCA plot (Figure 4a), which could indicate very distinct purge metabolome profiles observed with different aging days. Similarly, the heatmap also showed two distinct clusters in which 2 d purge samples aggregated together and significantly differed from 9 d and 16 d samples (Figure 4b). Although 9 d and 16 d samples presented similar metabolome profiles, these samples still produced some distinct clusters, respectively. The purge metabolome from loin 9 showed some differences from the other metabolomes, which could be due to differences in the biological samples themselves. Basically, since the metabolome profiles of pork purge were significantly affected by aging treatments, it is worthwhile to emphasize that purge from shorter aged (2 d postmortem aged) pork presented totally different metabolome profiles than did longer aged pork purge (9 d and 16 d).

A total of 933 features changed significantly in three pairwise comparisons (16 d vs. 2 d, 9 d vs. 2 d, and 16 d vs. 9 d) (Figure 4c) with a fold change of 2 (ratio > 2 or <0.5) and a false discovery rate (FDR) < 0.05, in which 34 features were tentatively identified via matching METLIN database according to MS/MS information (Table 2).

Ten compounds were overabundant in 2 d postmortem aged purge (Table 2), and these compounds were involved in ATP synthesis and metabolism (creatine, hypoxanthine), histidine metabolism (L-Histidine, carnosine), phenylalanine metabolism (2-Phenylacetamide), and the antioxidation of reactive oxygen species (ROS) and free radicals (glutathione, oxidized GSSH). Postmortem ATP synthesis and metabolism radically determines most of the critical qualities of raw meat. Creatine has the ability to increase muscle stores of phosphocreatine, potentially increasing the muscle’s ability to resynthesize ATP from ADP in order to meet increased energy demands [39]. The muscle ATP concentration remains stable through the phosphocreatine shuttle early in the postmortem process. However, muscle stores of phosphocreatine are also limited (around 25 mmol/g of muscle tissue) and the phosphocreatine buffer system can only sustain postmortem cellular ATP for a brief period [26], even though creatine and phosphocreatine could be still detected in certain concentrations in pork meat after 44 d of aging [40]. The decline of creatine content in pork purge during aging in the present study coincided with the decreased tendency of creatine and phosphocreatine levels in pork meat after longer aging treatments [40]. This might suggest a progressive collapse of the phosphocreatine shuttle during aging. Meanwhile, hypoxanthine (a degradation product of ATP) also declined in pork purge during aging. It was reported that the ATP content will rapidly decrease and almost be exhausted at 24 h postmortem in pork longissimus lumborum (LL) muscles, followed by an increase in the downstream product of ATP degradation [41]. However, hypoxanthine can be oxidized into xanthine via xanthine dehydrogenase; thus, it could be decreased in pork purge alongside aging.

When animals are slaughtered for meat, oxygen deficiency results in the impaired mitochondrial respiration, producing ROS [40]. Glutathione and its redox forms (oxidized glutathione as GSSH, reduced glutathione as GHS) plays a critical role in preventing damage caused by ROS (such as free radicals, peroxides, lipid peroxides, and heavy metals) [42] for cellular components. GSSG is produced when peroxides are detoxified by GSH peroxidase and are recycled back to their reduced form by GSH reductase, at the expense of nicotinamide adenine dinucleotide phosphate (NADPH). The significantly lower level of GSSG in 16 d aged purge samples as compared to that in 2 d aged samples might indicate the gradually incapacitated antioxidant defense system in pork muscle during the aging period. Meanwhile, carnosine and L-histidine showed significant lower levels in 16 d aged purge compared to those in 2 d aged samples. Carnosine, a dipeptide composed of L-histidine and β-alanine, is a potential endogenous antioxidant that scavenges ROS as well as inhibits lipid oxidation during oxidative stress [43,44,45]. L-histidine has also been demonstrated to have an antioxidative action and has proved to be an excellent scavenger of reactive ^1^O_2_ species [46]. Taken together, the decrease in GSSG carnosine and L-histidine identified in pork purge during aging would be likely associated with increased lipid and protein oxidation (Figure 3) in pork meat.

At the same time, 24 compounds were identified in overabundance in 9 d and 16 d aged pork purge (longer aging treatments). Basically, most of these compounds were dipeptides and tripeptides. The significantly increased levels of these peptides in purge after 9 d of aging can be explained by accumulated proteolysis, which starts after slaughter. The postmortem proteolysis by several endogenous proteolytic enzyme systems (calpains, cathepsins, and proteasomes) is mainly responsible for the degradation of structural proteins [47], which are broken down into polypeptide fragments, inducing an increased tenderness of the meat (reflected by a declined shear force and an increased MFI, as seen in Figure 2). This is followed by peptidyl peptidases acting to generate small peptides. The aforementioned changes induce a relative significantly increased concentration of peptides in the muscle with a longer postmortem aging time [48,49]. Thus, it is speculated that increasing the amount of peptides effused from muscle with water loss and detecting them in purge as aging indicators (Table 2), could make them useful as potential biomarkers for predicting meat tenderness during aging.

## 4. Conclusions

In summary, the present study indicated that aging significantly affected meat quality attributes as well as meat exudate metabolome profiles. Metabolites associated with ATP synthesis and metabolism (creatine and hypoxanthine), antioxidation (GSSG and carnosine), and proteolysis (dipeptides and tripeptides) could act as potential biomarkers to monitor aging times and indicate meat quality changes, such as increased lipid and protein oxidation, discoloration, and tenderness during aging processing. Further studies regarding target compound (potential biomarkers) quantification and proteome profiles of meat exudate would be highly warranted to provide more valuable information associated with meat quality.

## Figures and Tables

**Figure 1 foods-10-00668-f001:**
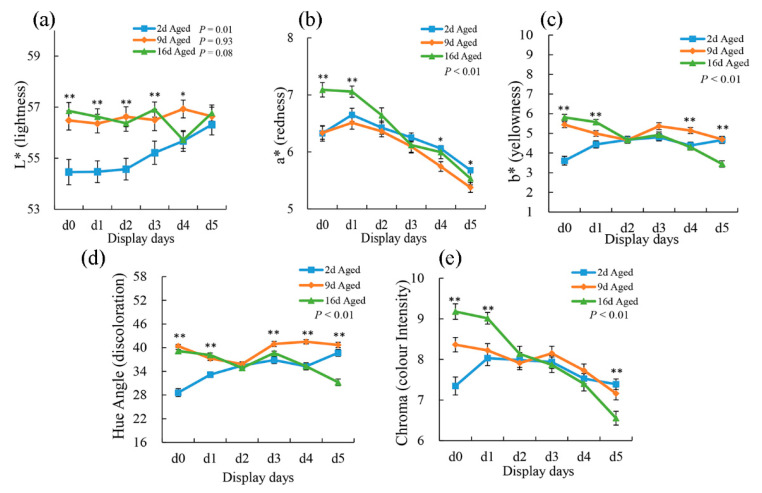
The effect of aging on the changes in L* (**a**), a* (**b**), b* (**c**), hue angle (**d**), and chroma (**e**) of pork steak during 5 days of display. The results are expressed as the mean ± standard error. The significant differences among three aging treatments at the same display time were marked with asterisks (* means *p* < 0.05; ** means *p* < 0.01).

**Figure 2 foods-10-00668-f002:**
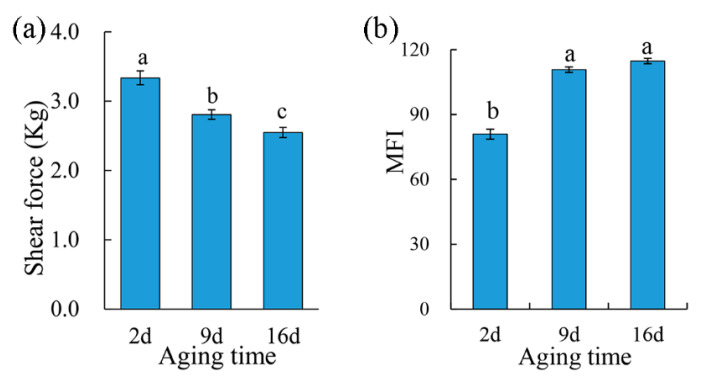
The effect of aging treatments on shear force (**a**) and the myofibril fragmentation index (**b**) of the pork meat. Results are expressed as the mean ± standard error. Means with different letters (a–c) are different (*p* < 0.05).

**Figure 3 foods-10-00668-f003:**
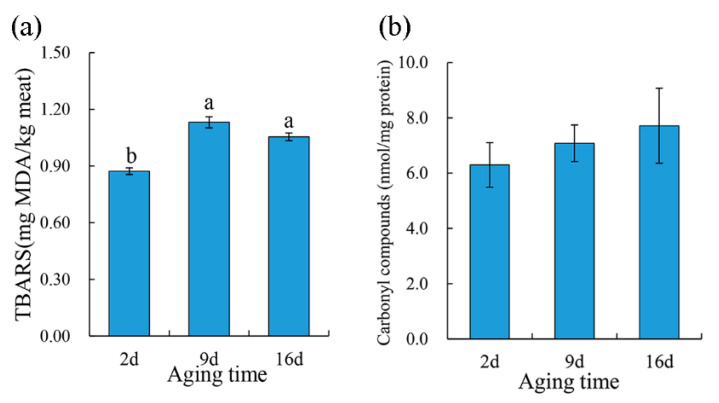
The effect of aging treatments on lipid (**a**) and protein (**b**) oxidation of pork meat. Results are expressed as the mean ± standard error. Means with different letters (a, b) are different (*p* < 0.05).

**Figure 4 foods-10-00668-f004:**
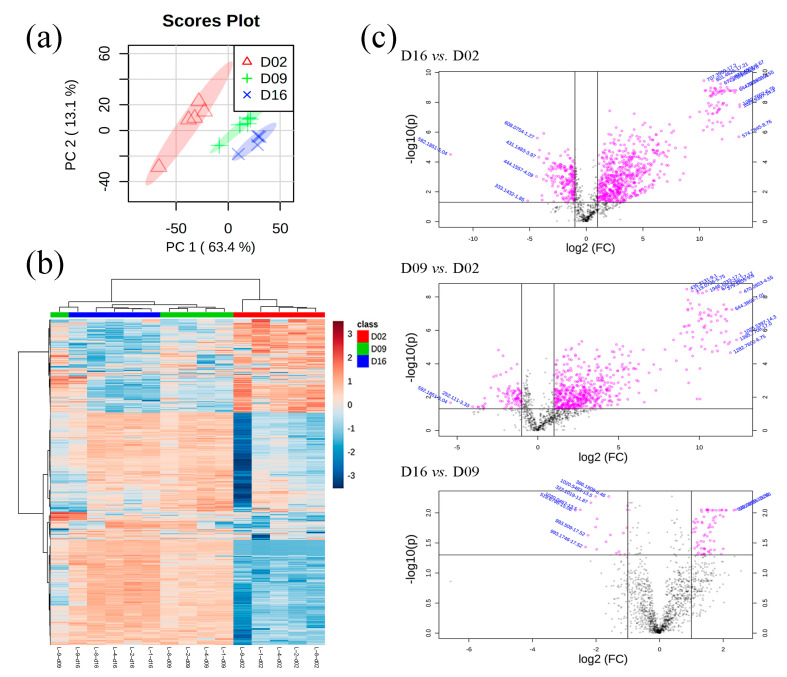
The principal component analysis (PCA) score plots (**a**), hierarchical cluster analysis (HCA) score plots (**b**), and volcano plots (**c**) of metabolome in pork exudate. D02, D09, and D16 refer to pork exudate from 2 d, 9 d, and 16 d postmortem aging treatments, respectively.

**Table 1 foods-10-00668-t001:** The pH and water-holding capacity changes of pork loins during postmortem aging.

Aging Time	1 d (24 h)	2 d	9 d	16 d
pH	5.65 ± 0.01 ^a^	5.55 ± 0.01 ^c^	5.59 ± 0.01 ^b^	5.57 ± 0.01 ^b^
Purge loss (%)	-	3.23 ± 0.30 ^c^	5.41 ± 0.50 ^b^	7.80 ± 0.55 ^a^
Drip loss (%)	-	6.17 ± 0.31 ^a^	2.74 ± 0.16 ^b^	1.46 ± 0.18 ^c^
Cooking loss (%)	-	28.33 ± 0.90	28.01 ± 0.73	25.74 ± 1.06

Results are expressed as the mean ± standard error. Means with different letters (^a^–^c^) in a row are different (*p* < 0.05).

**Table 2 foods-10-00668-t002:** Tentatively identified metabolites that have changed significantly among the three aging treatments.

Name	Mass	*m/z*	MS1 Error (ppm)	RT ^1^	Adduct Ion Name	D16 vs. D02		D09 vs. D02		D16 vs. D09	
						Fold-Change	FDR ^2^	Fold-Change	FDR	Fold-Change	FDR
**Compounds overabundant in 2 d postmortem aging (12)**											
Creatine	131.0703	132.0774	−5.15	0.92	(M+H)^+^	0.457	0.026	0.372	0.004		
2-Phenylacetamide	135.0795	136.0764	−5.08	3.13	(M+H)^+^	0.344	0.019	0.339	0.006		
Hypoxanthine	136.0382	137.0466	−5.75	1.32	(M+H)^+^	0.127	0.004	0.238	0.048		
N-Methylisoleucine	145.1103	146.1183	−4.81	4.31	(M+H)^+^	0.426	0.001				
L-Histidine	155.0698	156.0773	−3.2	1.04	(M+H)^+^	0.473	0.000				
Carnosine	226.1071	227.1149	−4.69	1.03	(M+H)^+^	0.420	0.011				
Isobutyryl carnitine	232.1432	232.1559	−6.88	4.75	(M)^+^	0.423	0.010				
His Glu	284.1101	285.1208	−5.29	1.36	(M+H)^+^	0.413	0.000				
Lys Pro Ile	356.2432	357.2509	−3.5	0.83	(M+H)^+^	0.492	0.002				
Glutathione, oxidized	612.1505	613.1613	−3.39	1.49	(M+H)^+^	0.432	0.000				
**Compounds overabundant in 9 d and 16 d postmortem aging (27)**											
Pro Ala	186.1004	187.1086	−4.88	1.12	(M+H)^+^	3.170	0.001	2.206	0.006		
Ala Val	188.1155	189.1242	−4.23	1.32	(M+H)^+^	2.175	0.000				
Ile Gly	188.1162	189.1242	−4.23	2.86	(M+H)^+^	3.190	0.000	2.198	0.000		
L-Leucyl-L- Alanine	202.1319	203.14	−4.93	2.53	(M+H)^+^	3.316	0.003	3.011	0.009		
Spermine	202.1317	203.2239	−4.12	3.28	(M+H)^+^	2.899	0.018	2.480	0.042		
Val Ser	204.111	205.1196	−6.27	0.88	(M+H)^+^	2.766	0.001	2.018	0.011		
Pro Thr	216.1095	217.1191	−3.88	1.25	(M+H)^+^	4.626	0.001	3.379	0.004		
Val Val	216.1477	217.1556	−4.14	3.43	(M+H)^+^	7.910	0.000	5.018	0.000		
Ile Ser	218.1267	219.1347	−3.7	1.2	(M+H)^+^	4.945	0.000	3.137	0.002		
Val Leu	230.1631	231.1711	−3.57	5.69	(M+H)^+^	4.716	0.000	3.693	0.003		
Asn Val	231.1215	232.1303	−4.81	1.3	(M+H)^+^	4.741	0.000	4.416	0.012		
Leu Thr	232.1424	233.1506	−4.45	1.9	(M+H)^+^	5.627	0.003	3.262	0.042		
Ala Phe	236.1162	237.1244	−4.24	5.76	(M+H)^+^	8.885	0.000	4.650	0.007		
Ile Ile	244.1783	245.1869	−3.77	7.8	(M+H)^+^	6.780	0.001	3.081	0.014		
Ile Asp	246.1218	247.1296	−3.2	4.05	(M+H)^+^	2.105	0.000	2.146	0.000		
Asp Leu	246.122	247.1296	−3.2	4.43	(M+H)^+^	5.903	0.000	4.222	0.000		
Tyr Ala	252.1115	253.1196	−5.21	2.42	(M+H)^+^	2.381	0.026	2.911	0.027		
Ala Ile Gly	259.1553	260.1617	−4.61	1.36	(M+H)^+^	6.617	0.000	5.237	0.000		
Pro Leu Gly	285.1685	286.1771	−3.26	6.11	(M+H)^+^	3.141	0.007	2.213	0.025		
Val Ile Gly	287.186	288.1939	−7.4	4.32	(M+H)^+^	9.503	0.000	4.396	0.007		
His Phe	302.1376	303.1463	−3.65	2.74	(M+H)^+^	9.624	0.000	5.950	0.001		
Gly Lys Leu	316.2111	317.2195	−3.54	2	(M+H)^+^	123.990	0.000	37.504	0.001	3.306	0.041
Val Val Asp	331.1744	332.1832	−4.66	3.51	(M+H)^+^	47.939	0.000	19.345	0.001		
Val Glu Glu	375.1641	376.173	−4.06	3.07	(M+H)^+^	3.433	0.005	3.348	0.024		

^1^ RT is the average retention time. ^2^ FDR is the false discovery rate of the T-test. D02, D09, and D16 refer to pork exudate from 2 d, 9 d, and 16 d postmortem aging treatments, respectively.

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
