# Peer review of "Utilizing Pork Exudate Metabolomics to Reveal the Impact of Aging on Meat Quality"

_foods, 2021, doi:10.3390/foods10030668_

Round 1

Reviewer 1 Report

In this manuscript, the authors perform metabolomic profiling of meat exudates with different days of aging. Overall, I think this topic is interesting. Currently, not many of meat exudate metabolomics studies have been published. The manuscript is well written, but the authors should check and correct some typos. The manuscript is also well organized, and the authors provide enough information in experimental part. Here are some comments below:

  • For metabolite extraction, chloroform could remove most of the lipids. However, there will be a risk of losing other non-polar metabolites. Could the authors explain why they use this two-layer extraction method?
  • Did the authors inject quality control (QC) samples to check the stability of LC-MS? QC samples’ reproducibility is very important in metabolomics research. If QC samples are not clustered in PCA plot, then the metabolomics dataset itself is meaningless.
  • The authors should improve figure quality. There are too many labels on PCA plot, and the numbers on x- and y-axis of volcano plots are too small.
  • Line 68-69, “relationship between identified metabolites and oxidative stability of beef muscles during aging were conducted as well”, should be “relationship between identified metabolites and oxidative stability of beef muscles during aging was conducted as well”
  • Line 127, “Potassium phosphate” should be “potassium phosphate”.
  • The authors only applied positive ion mode scan in MS analysis. However, most of carboxylic acids can be only analyzed by negative ion mode. Could the authors explain why they only used the positive ion scan mode?
  • The definition of small molecule is the compound with molecular weight less than 1000 Da. Why the authors selected the mass ranges from 70-2000?
  • Line 182, what is the definition of small value here? Is it the minimum value of each variable?
  • Line 258, should be “shear force”
  • The authors may consider performing a pathway enrichment analysis. This will provide the authors a global view of impacts of aging on metabolic pathways.

Author Response

In this manuscript, the authors perform metabolomic profiling of meat exudates with different days of aging. Overall, I think this topic is interesting. Currently, not many of meat exudate metabolomics studies have been published. The manuscript is well written, but the authors should check and correct some typos. The manuscript is also well organized, and the authors provide enough information in experimental part. Here are some comments below:

  • For metabolite extraction, chloroform could remove most of the lipids. However, there will be a risk of losing other non-polar metabolites. Could the authors explain why they use this two-layer extraction method?

Our lab has standardized on using the ubiquitous Bligh-Dyer extraction protocol because it allows metabolites and lipids to be fractionated from the same sample, as well as effectively removing protein. The water/methanol phase contains the polar and mid-polarity metabolites. Our HPLC analysis was performed using a Waters T3 column. This is not a conventional C18 reversed-phase column. The T3 is a modified phase that gives higher retention for polar molecules. The emphasis for the study presented in this manuscript was a focus on more polar metabolites. It is true that some non-polar metabolites may partition into the chloroform layer, which was not analyzed in this study, and not be observed. 

Reference:

Snyder, N.W., Khezam, M., Mesaros, C.A., Worth, A., Blair, I.A. Untargeted Metabolomics from Biological Sources Using Ultraperformance Liquid Chromatography-High Resolution Mass Spectrometry (UPLC-HRMS). J. Vis. Exp. (75), e50433, doi:10.3791/50433 (2013).

  • Did the authors inject quality control (QC) samples to check the stability of LC-MS? QC samples’ reproducibility is very important in metabolomics research. If QC samples are not clustered in PCA plot, then the metabolomics dataset itself is meaningless.

Given the relatively low number of samples used in this study, QCs were not periodically inserted within the sample set. However, samples were randomized to address for potential instrumental bias. 

  • The authors should improve figure quality. There are too many labels on PCA plot, and the numbers on x- and y-axis of volcano plots are too small.

Thank you so much for your suggestion. The Figure 4 has been improved accordingly, please find in revised manuscript.

  • Line 68-69, “relationship between identified metabolites and oxidative stability of beef muscles during aging were conducted as well”, should be “relationship between identified metabolites and oxidative stability of beef muscles during aging was conducted as well”

It has been revised accordingly, please check line 68-69.

  • Line 127, “Potassium phosphate” should be “potassium phosphate”.

The manuscript was revised accordingly. Please check line 127.

  • The authors only applied positive ion mode scan in MS analysis. However, most of carboxylic acids can be only analyzed by negative ion mode. Could the authors explain why they only used the positive ion scan mode?

The authors appreciate the reviewer’s point. Some chemical classes offer better instrument sensitivity in positive mode and some are better in negative mode.  Having assayed the samples in both positive and negative modes would result in more complete coverage of the metabolome.  However, in our experience, carboxylic acids often source fragment when the instrument is operated in negative mode, leading to false or missed compound identification via parent ion database matching.

  • The definition of small molecule is the compound with molecular weight less than 1000 Da. Why the authors selected the mass ranges from 70-2000?

Line 174 was written in error. The mass data was collected between 70 – 1000 m/z. This has been corrected. Please check line 174.

  • Line 182, what is the definition of small value here? Is it the minimum value of each variable?

The small value is defined by the half of the minimum values in the original data. We have added the additional information, please check line 182-183.

  • Line 258, should be “shear force”

The manuscript was revised accordingly. Please check line 258.

  • The authors may consider performing a pathway enrichment analysis. This will provide the authors a global view of impacts of aging on metabolic pathways.

Thank you for your suggestion. We agree that pathway enrichment analysis can give much valuable information especially for dealing with large number of significant metabolites. The significant enriched metabolic pathway could be extracted related to the differentially abundant metabolites by the pathway enrichment analysis. However, in our present research, only roughly 30 compounds were tentatively identified, and most of them were dipeptides and tripeptides. Most of peptides cannot be hit in KEGG pathway searches. Thus, we did not perform pathway enrichment analysis.

Reviewer 2 Report

The authors investigated "Metabolomics profiling to reveal the impact of aging on quality attributes of pork loins". The manuscript is basically suitable for the purpose of journal. However, some modifications are needed.

1)The quality of Table 2 is too bad to appeal the results. This should be more sophisticated. 

2)Not only non-targeted but also targeted analysis of metabolites should be done. By the quantification of metabolites, readers can believe the results.

3)Table 2 is very difficult to understand. Authors must select one ion for one metabolite based on the quality of the peak. Readers can not judge which is a better annotation for the metabolites (for example, L-histidine, 2-phenilacetamide, Ile Ser. If they can not, annotation quality should be added for each metabolite.

4)Relationship between important phenotype and metabolite should be analyzed more clearly. For example, the relationship between TBARS and metabolite should be statistically analyzed by not only multivariate but also univariate analysis.

Author Response

The authors investigated "Metabolomics profiling to reveal the impact of aging on quality attributes of pork loins". The manuscript is basically suitable for the purpose of journal. However, some modifications are needed.

1)The quality of Table 2 is too bad to appeal the results. This should be more sophisticated.

The authors appreciate the reviewer’s comment and suggestion. While presentation of analyzed metabolomics data can be provided in several different ways (and possibly with more complicated features), the present table should provide sufficient information to readers in our discipline to get some essential/initial information responding to the primary research question/objective of the current study.

2)Not only non-targeted but also targeted analysis of metabolites should be done. By the quantification of metabolites, readers can believe the results.

Non-targeted profiling is an initial screening or discovery study. We agree that future work should validate the tentatively identified compounds by performing a follow-up targeted analysis study, such as using HPLC-triple quadrupole mass spectrometry or targeted quantification of each specific metabolite. In the present study, most of the metabolites profiled were dipeptides and tripeptides.

3)Table 2 is very difficult to understand. Authors must select one ion for one metabolite based on the quality of the peak. Readers cannot judge which is a better annotation for the metabolites (for example, L-histidine, 2-phenilacetamide, Ile Ser. If they cannot, annotation quality should be added for each metabolite.

Thank you for your suggestion. We agree with your point. We have been checked it again and selected one ion for one metabolite based on the score and quality of peak. Please find the revised Table 2 in manuscript.

4)Relationship between important phenotype and metabolite should be analyzed more clearly. For example, the relationship between TBARS and metabolite should be statistically analyzed by not only multivariate but also univariate analysis.

Thank you for your suggestion. As the reviewer is aware, untargeted metabolomics can provide thousands of compounds, multivariate analyses, such as PCA, HCA, (O)PLS-DA, are effective methods to find the underlying variation tendency of thousands of compounds. In our present study, a total of 1483 features were detected, quite a few of them showed increased tendency along with aging time, as we can see from heatmap of HCA. TBARS, carbonyl content, and MFI value also showed similar trends, and if we do the Pearson correlation analysis or univariate analysis, we may get lots of compounds that correlated with TBARS, but maybe a lot of them do not have actual relationship with lipid oxidation, such as peptides (peptides should relate with accumulated proteolysis and MFI). We believe that meat quality changes are not determined by a single metabolite. Further, specific targeted quantification would be also followed to verify the accuracy of actual correlation. Thus, multivariate analyses are relatively suitable methods to reveal a global view of aging on meat metabolome changes.